# Aerobic Exercise Improves Radiation Therapy Efficacy in Non-Small Cell Lung Cancer: Preclinical Study Using a Xenograft Mouse Model

**DOI:** 10.3390/ijms25052757

**Published:** 2024-02-27

**Authors:** Sunmi Jo, Jaewan Jeon, Geumju Park, Hwan-Kwon Do, JiHoon Kang, Ki Jung Ahn, Sun Young Ma, Young Min Choi, Donghyun Kim, BuHyun Youn, Yongkan Ki

**Affiliations:** 1Department of Radiation Oncology, Haeundae Paik Hospital, Inje University School of Medicine, Busan 48108, Republic of Korea; smjo@paik.ac.kr (S.J.); jjw1066@paik.ac.kr (J.J.); geumju.park@paik.ac.kr (G.P.); 2Department of Physical Medicine and Rehabilitation, Haeundae Paik Hospital, Inje University School of Medicine, Busan 48108, Republic of Korea; satirev83@paik.ac.kr; 3Department of Hematology and Medical Oncology, Winship Cancer Institute of Emory, Emory University School of Medicine, Atlanta, GA 30322, USA; jhkang4293@gmail.com; 4Department of Radiation Oncology, Busan Paik Hospital, Inje University School of Medicine, Busan 48108, Republic of Korea; 103803@paik.ac.kr; 5Department of Radiation Oncology, Kosin University Gospel Hospital, Kosin University College of Medicine, Busan 49267, Republic of Korea; moomsy@hanmail.net; 6Department of Radiation Oncology, Dong-A University College of Medicine, Busan 49315, Republic of Korea; cymin00@dau.ac.kr; 7Department of Radiation Oncology and Biomedical Research Institute, Pusan National University School of Medicine, Busan 49241, Republic of Korea; dh2372@daum.net; 8Department of Biological Sciences, Pusan National University, Busan 46241, Republic of Korea; 9Department of Radiation Oncology, Pusan National University Yangsan Hospital, Pusan National University School of Medicine, Yangsan 50612, Republic of Korea

**Keywords:** aerobic exercise, radiation therapy, lung cancer, NSCLC, oxygenation

## Abstract

The “oxygen effect” improves radiation efficacy; thus, tumor cell oxygen concentration is a crucial factor for improving lung cancer treatment. In the current study, we aimed to identify aerobic exercise-induced changes in oxygen concentrations in non-small cell lung cancer (NSCLC) cells. To this end, an NSCLC xenograft mouse model was established using human A549 cells. Animals were subsequently subjected to aerobic exercise and radiation three times per week for 2 weeks. Aerobic exercise was performed at a speed of 8.0 m/m for 30 min, and the tumor was irradiated with 2 Gy of 6 MV X-rays (total radiation dose 12 Gy). Combined aerobic exercise and radiation reduced NSCLC cell growth. In addition, the positive effect of aerobic exercise on radiation efficacy through oxygenation of tumor cells was confirmed based on hypoxia-inducible factor-1 and carbonic anhydrase IX expression. Finally, whole-transcriptome analysis revealed the key factors that induce oxygenation in NSCLC cells when aerobic exercise was combined with radiation. Taken together, these results indicate that aerobic exercise improves the effectiveness of radiation in the treatment of NSCLC. This preclinical study provides a basis for the clinical application of aerobic exercise to patients with NSCLC undergoing radiation therapy.

## 1. Introduction

Lung cancer is the leading cause of cancer-related deaths worldwide, accounting for the highest mortality rate in men and women [1,2]. According to the Global Cancer Observatory, lung cancer incidence and mortality rates are similar across 20 countries with igh incidences of lung cancer [3]. Environmental factors, such as smoking and pollution, and genetic factors can cause lung cancer; however, smoking remains the leading cause [4,5]. Small cell lung cancer and non-small cell lung cancer (NSCLC) are the two most common types of lung cancer, with NSCLC comprising ~85% of lung cancer cases [6,7].

Treatment options for lung cancer include surgery, chemotherapy, and radiation therapy. However, stereotactic body radiation therapy and image-guided radiation therapy have emerged as effective treatment strategies [8]. To date, numerous anticancer drugs have been developed to treat lung cancer [9,10,11], leading to significantly improved early stage NSCLC prognosis. However, the treatment of advanced lung cancer remains challenging.

To improve lung cancer treatment efficacy, radiation therapy is combined with various anticancer drugs [12,13,14]. However, combination therapy is associated with many side effects, including oral mucositis, gastrointestinal toxicity, and hematopoietic damage, which hinder continuous clinical treatment [15,16]. Another method used to increase radiation therapy efficacy is tumor cell reoxygenation [17]. In 1950, Gray and Thomlinson evaluated the relationship between tumor radiation response and oxygen and explored the potential implications of the “oxygen effect” [18,19]. Gray and Thomlinson’s histological studies of human lung cancer and mathematical modeling of intra-tumoral oxygen transport, oxygen consumption, and regional O_2_ distribution within the tumor suggested that oxygen is closely related to tumor survival [20]. These findings promoted the study of radiobiological oxygen and hypoxia effects on cancer treatment [21,22,23,24]. Tumor hypoxia is a major contributing factor to the failure of radiation therapy, while the “oxygen effect” is essential for increasing radiation therapy efficacy [25]. Accordingly, recent research has focused on increasing oxygen levels in cancer cells through the direct delivery of oxygen using nanoparticles [26]. To achieve superior penetration into tumor cells, catalytic nano-shuttles have been developed [27,28]. Despite significant technological improvements in oxygen delivery methods, there is an ongoing debate regarding the choice of nano-shuttle materials and nanoparticle sizes for optimized tumor treatment. In addition, from a safety perspective, challenges persist regarding the design of a system that ensures the particles can travel through blood vessels to specifically and directly target tumor cells.

Several studies have shown that aerobic exercise can regulate tumor vasculature and oxygenation, leading to the development of relatively normal, more mature, and less permeable vessels in dysfunctional tumor vasculature. Aerobic exercise can also significantly increase intratumoral vascularization, leading to the normalization of the tissue microenvironment in human breast tumors [29]. Such findings may have important implications for inhibiting tumor metastasis and improving the study of the role of physical activity and exercise across the entire cancer continuum [30]. However, the mechanisms by which aerobic exercise inhibits or delays tumorigenesis currently remain elusive, and the study of the relationship between aerobic exercise and tumors requires further investigation to develop more effective clinical treatment regimens [29,30,31]. Aerobic exercise is an alternative method to induce the “oxygen effect” within tumor cells, although clinical studies have shown that it elicits variable effects on oxygen saturation [32,33,34].

The current preclinical study investigates the effect of aerobic exercise on oxygen changes in tumor cells to enhance the effectiveness of radiation therapy. In addition, whole mRNA sequencing was performed on tumor tissues to identify key factors contributing to the “oxygen effect” when aerobic exercise and radiation are combined.

## 2. Results

### 2.1. Combined Aerobic Exercise and Radiation Therapy Reduces NSCLC Cell Growth

We investigated the effect of combined aerobic exercise and radiation therapy on NSCLC cell growth utilizing an in vivo tumor xenograft model (Figure 1A). At 4 weeks, decreased tumor volume was observed in the radiation-alone group and the combination group. However, there was no significant difference in tumor volume reduction between the combination and radiation-alone groups (Figure 1B,C). Hematoxylin and eosin (H & E) staining and Kiel 67 (Ki-67) immunohistochemistry (IHC) were performed to assess the growth and proliferation levels of NSCLC cells in mouse tumor tissue sections. H&E staining revealed significant differences in cell damage between the radiation-alone and complex groups; however, KI-67 staining revealed no significant differences in cell proliferation (Figure 1D).

### 2.2. Aerobic Exercise Combined with Radiation Induces Changes in Tumor Cell Oxygen Concentration

Oxygen distribution in tumor cells is an important factor for increasing radiation efficacy. To confirm whether aerobic exercise induces the “oxygen effect” in lung cancer cells, IHC was performed using hypoxia-inducible factor-1 α (HIF-1α) and carbonic anhydrase IX (CA IX) antibodies on mouse tumor tissue sections (Figure 2A,B).

The expression of HIF-1α and CA IX in the combination group was significantly higher than in the control and radiation-alone groups. In addition, the reactive oxygen species (ROS) levels of the combination group were higher than those of the control group (Figure 2C). Consistently, oxygen concentration and ROS levels were higher in the combination group than in the radiation-alone group.

### 2.3. Aerobic Exercise Combined with Radiation Alters Gene Expression

To evaluate the effects of aerobic exercise on gene expression, whole transcriptome analysis was conducted using tissues obtained from the in vivo experiments. The Gene Expression Omnibus accession number of the data was GSE221562. We identified differentially expressed genes (DEGs) that were upregulated or downregulated by at least two-fold relative to gene expression levels in controls, using comparative combination analysis. When aerobic exercise and radiation were combined, 179 and 156 genes were upregulated and downregulated, respectively (Figure 3A). The top 20 DEGs are summarized in Table 1. Additionally, we identified 215 DEGs between the radiation-alone group and the combination group (Figure 3B).

To determine the functions of the altered DEGs, gene ontology (GO) enrichment (biological process [BP] and molecular function [MF] of individual genes and their cellular components [CC]) and Kyoto Encyclopedia of Genes and Genomes (KEGG) analyses were performed. In the GO_BP analysis, the GO terms significantly altered by aerobic exercise were primarily involved in the regulation of vascular development and angiogenesis (Figure 3C). For the GO_CC analysis, the GO terms that were significantly altered by aerobic exercise were protein-DNA complexes, DNA-packing complexes, and nucleosomes (Figure 3D). In addition, aerobic exercise altered cancer-associated microRNAs (Figure 3E).

### 2.4. DEGs That Alter the Oxidation Levels of NSCLC Cells during Combined Treatment

We investigated key factors that induced changes in the oxygen concentration of tumor cells in the combination group compared to the radiation-alone group. We also investigated the DEGs involved in the changes in ROS levels in tumor cells when aerobic exercise and radiation were administered.

A notable difference between the radiation-alone group and the combination group was observable in the overall DEG heat map in four groups related to ROS and oxygenation. Of these, we further investigated four genes (glutathione S-transferases [Gst] Mu [*Gstm*] 5 [*Gstm5*], mitochondrial permeability transition pore [*Mptp*], GST omega 1 [*Gsto1*], and solute carrier family 25 member 31 [*Slc25a31*]). Further, KEGG analysis confirmed that these genes were involved in the ROS pathway (Figure 4B).

According to the constructed database, the *Gstm5* expression level increased by 4.75-fold, *Slc25a31* gene expression increased by 1.96-fold, and *Gsto1* expression increased by 5.88-fold in the combination group compared to the radiation-alone group. Finally, the expression of *Mptp* increased by 3.65-fold in the aerobic exercise group compared to the combination group.

## 3. Discussion

Currently, radiation therapy plays an important role in the treatment of NSCLC [35], and tumor oxygenation and reoxygenation during radiation therapy are important in predicting NSCLC responses [36]. Accordingly, many studies have used nanoparticles to directly deliver oxygen to tumor cells [37,38]. Despite advances in nano-carrier formulation technology, clinical application of these technologies for the treatment of patients with lung cancer remains limited. In this study, aerobic exercise was considered an alternative method to deliver oxygen to tumors. Some studies have shown that oxygen saturation decreases after aerobic exercise; however, others have highlighted the positive aspects of aerobic exercise for the treatment of other diseases, including cancers. For example, aerobic exercise has been used as an adjuvant treatment for patients with cancer, including those with breast or prostate cancer [39]. Aerobic exercise has also been used as an effective adjuvant therapy to improve peripheral oxygen saturation in patients with diabetes [40]. In addition, in patients with chronic obstructive pulmonary disease, 6 weeks of aerobic exercise improved oxygen saturation and submaximal performance [41]. Interestingly, rectal toxicity decreased with aerobic exercise in patients with prostate cancer who received radiation therapy [42].

HIF-1α and CA IX were used as markers to confirm changes in oxygen concentration in NSCLC cells when aerobic exercise and radiation were combined. HIF-1 is a transcription factor that belongs to the basic helix-loop-helix group. HIF-1α and HIF-1β form a heterodimer and induce the expression of several genes, including the gene encoding vascular endothelial growth factor, involved in angiogenesis [43]. HIF-1α activity is regulated by intracellular oxygen concentration [44]. CA IX, like HIF-1α, is a major marker of hypoxic state in tumor cells and is used in tumor cell prognosis [45]. CA IX levels are significantly increased by tumor cell oxygenation [46]. Additionally, CA IX expression leads to restoration of the S/G2 phase in reoxygenated cells [47]. In preclinical animal experiment results, the expression of these two markers was increased in the combination group than in the radiation-alone group.

We also investigated ROS changes in tumor cells when aerobic exercise was combined with radiation. ROS act as oxygen sensors in tumor cells and are key factors in increasing the effectiveness of radiation therapy [48]. In addition, ROS are closely related to tumor cell development, growth, metastasis, and apoptosis [49,50]. Nicotinamide adenine dinucleotide phosphate hydrogen oxidase 4 and H_2_O_2_ increase HIF-1α transcription, indicating that HIF-1α expression is increased by activating the nuclear factor kappa-light-chain-enhancer of activated B cells signaling pathway by ROS [51,52]. According to Eleftheriadis et al., reoxygenation induces ROS by activating aryl hydrocarbon receptors and interacting with HIF-1α [53]. Our animal experiments revealed that HIF-1α and ROS levels were increased in the combination group compared to the radiation-alone group.

Additionally, we investigated genes that showed altered expression when aerobic exercise was combined with radiation. DEG analysis revealed that microRNA (miR)-106B expression was significantly increased in the combination group compared to the control group. In a recent study, miR-106B was found to be under-expressed in colorectal cancer (CRC), with miR-106B downregulation negatively associated with lymph node metastasis. MiR-106B has also been shown to inhibit CRC metastasis by regulating cathepsin A expression [54]. In this study, we focused on genes associated with oxygenation and ROS in the combination group; four of the altered DEGs were of interest. *Gstm* is the largest group in the GST family and comprises *Gstm1*–*m5* according to the similarity of amino acid sequences [55]. According to Leonard et al., reoxygenation induced in the renal tissue following hypoxia-induced ischemia-reperfusion injury upregulates antioxidant genes, including *Gstm5* [56]. These results indicate that reoxygenation activity is involved in regulating cell recovery and apoptosis under hypoxic conditions and that ROS may play an important role in this process. In addition, *GSTM5* is expressed at low levels in human bladder cancer cells and tumor cell growth is inhibited when *GSTM5* is overexpressed [57]. Interestingly, *GSTM5* represents a robust predictor of the overall survival of patients with NSCLC [58]. Our analysis showed that *Gstm5* expression increased by 4.78-fold in the combination group compared to the radiation-alone group.

MPTP is a pore located in the inner mitochondrial membrane that opens in response to oxidative stress and Ca^2+^ loading [59]. MPTP opening leads to mitochondrial expansion and release of the pro-apoptotic mediator cytochrome c, ultimately leading to apoptosis [60]. MPTP induces early reoxygenation during myocardial injury, suggesting that the formation of MPTP is key in myocardial injury recovery [61]. Additionally, various cancer cells, including those of cervical cancer and neuroblastoma, generate ROS through the MPTP pathway and ultimately lead to cancer cell death [62,63]. SLC25A31, a member of the adenosine nucleotide translocase (ANT) family, is a mitochondrial membrane protein involved in energy homeostasis and apoptosis [64]. Produced in the cytosol, SLC25A31 plays a role in adenosine diphosphate and adenosine triphosphate transport in and out of the mitochondrial matrix [65]. SLC25A31 is highly expressed in the human brain, liver, and testis [66]. Interestingly, recent studies have shown that ANT proteins induce necrotic cell death in the inner mitochondrial membrane by regulating MPTP activity [62]. In addition, ANT has been proposed as a new biomarker for the survival of patients with renal cancer; the cumulative survival probability decreases as ANT expression decreases [67]. In this study, we found that MPTP increased in the aerobic exercise-alone group compared to the combination group. That is, MPTP specifically increased only due to aerobic exercise and decreased when combined with radiation. Finally, *Gsto1* mRNA expression increased by 5.88-fold in the combination group compared to the radiation-alone group. GSTO1 induces interleukin 1B expression [68] and regulates ROS production through the Toll-like receptor 4 signaling pathway [69]. Xu et al. suggested that *GSTO1* is an influential cancer treatment target [70].

We conducted a preclinical study using an animal model to confirm the effectiveness of aerobic exercise as an oxygen supply method to increase the efficacy of radiation. However, our study was based only on a single mouse model, with five animals per group, and did not include NSCLC in vitro experiments. The primary focus of our investigation was to confirm the tendency of aerobic exercise on radiation therapy effects. In addition, the correlation between oxidative substances generated from aerobic exercise and radiation-induced ROS could not be elucidated. Clinical studies related to lung disease have shown that continuous aerobic exercise improves lung function and increases oxygen saturation [71,72]. However, given the difficulty associated with NSCLC patients undergoing radiation therapy performing continuous aerobic exercise, enrolment for clinical studies assessing the efficacy of this combined treatment strategy is low. Nevertheless, our study suggests that the effectiveness of radiation therapy can be improved through aerobic exercise, which is clinically easier to access than oxygen delivery using nanoparticles, and should be considered when performing radiation therapy for NSCLC. Further accumulation of clinical research data is needed, particularly in the establishment of a combination treatment protocol that delineates aerobic exercise duration and intensity, considering the individual status of patients receiving lung cancer radiation therapy.

## 4. Materials and Methods

### 4.1. Cell Culture

A549 cells, a human NSCLC cell line, were purchased from the Korea Cell Line Bank (Seoul, Republic of Korea). Cells were grown in Roswell Park Memorial Institute (RPMI) 1640 medium, supplemented with 10% fetal bovine serum, 100 U/mL penicillin, and 100 mg/mL streptomycin, at 37 °C and 5% CO_2_.

### 4.2. Animal Models

Six-week-old male Bagg Albino (BALB)/c athymic nude mice (Orient Bio, Seongnam, Republic of Korea) were used for the in vivo study. The animal experiment protocol used in this study was approved by the Pusan National University Institutional Animal Care and Use Committee (PNU-IACUC) for ethical procedures and scientific care (Approval Number PNU-2022-0060). The preclinical animal experiment consisted of four groups: control, radiation alone, aerobic exercise alone, and combination (aerobic exercise and radiation therapy). Five mice per group were placed in a sterile cage and maintained in an animal care facility in a temperature-regulated room (23 ± 1 °C) with a 12 h light/dark cycle and quarantined for 1 week prior to the study. The animals had access to water at all times and were fed a standard mouse chow diet.

The animals were injected with 5 × 10^5^ A549 cells, in the flank, and the tumors were allowed to develop. Upon identification of a palpable tumor (tumor volume: 200 mm^3^), the animals were subjected to combined radiation and aerobic exercise for 2 weeks. The mice were subjected to aerobic exercise before radiotherapy on the same day. Tumor diameter was measured with an electronic caliper and determined using the following formula: tumor volume was 4/3 πabc; for an oblate spheroid, a = b > c. Thus, tumor size was calculated as 4/3 πa2c, where a is the large diameter and c is the small diameter.

### 4.3. Histological Analysis

At the end of the treatment period, the animals were euthanized, and the tumors were fixed with formalin, dehydrated, embedded in paraffin, and sliced into 4–6 μm sections. The tumor sections were stained with H & E, according to standard protocols. Representative images of H & E staining for each group are shown in Figure 1 and Figure 2. To assess Ki-67, CA IX, and HIF-1α abundance via IHC, sections were treated with 3% hydrogen peroxide/methanol followed by 0.25% pepsin (S3002; Dako, Carpinteria, CA, USA) to retrieve antigens. Samples were then incubated in a blocking solution (X0909; Dako) and incubated overnight at 4 °C with primary antibodies in antibody diluent (S3022; Dako). Subsequently, the sections were washed with Tris-buffered saline with 0.1% Tween^®^ 20 detergent and incubated with a polymer-horseradish peroxidase-conjugated secondary antibody (K4001; Dako). A 3,3′-diaminobenzidine substrate chromogen system (K3468; Dako) was used to detect antibody binding. The stained sections were observed using a KFBio digital slide scanning system software (Konfoong Bioinformatics Technology Co., Ltd., Zhejiang, China, www.kfbiopathology.com).

For 4-hydroxynonenal and nitrotyrosine IHC, the optimal cutting temperature compound block was prepared as follows: the block was cut into 4 μm slices, rinsed with phosphate-buffered saline (PBS), and stained. A non-specific reaction was suppressed using a serum-blocking buffer, containing 0.5% bovine serum albumin. The sections were then incubated with a primary antibody overnight at 4 °C. Following incubation, the sections were rinsed thrice with PBS and incubated with fluorophore-conjugated secondary antibodies in a blocking buffer for an hour at room temperature in the dark. Finally, the sample was washed with PBS and mounted with 4′,6-diamidino-2-phenylindole-containing media.

### 4.4. Animal Radiation and Aerobic Exercise Treatment Procedure

As this was a preclinical study, five animals were assigned to each group to ensure statistical significance. Animals were exposed to X-rays using a linear accelerator (Varian Co., RapidArc, Palo Alto, CA, USA). The radiation energy used was a 6 MV X-ray, and the center of the tumor surface was set to 100 cm (source-to-skin distance). Aerobic exercises were performed using a treadmill controller (Columbus Instruments Co., Columbus, OH, USA). For radiation treatment, the mice were anesthetized and fixed in a radiation irradiator. The field size was adjusted so that radiation could be applied only to the flank area of the mice where the tumor was implanted. BALB/c athymic nude mice (n = 5 per group) were subjected to aerobic exercise for 30 min three times per week for 2 weeks, followed by radiation. Aerobic exercise was performed on a treadmill at a speed of 8.0 m/min, and the tumor was irradiated with 2 Gy radiation. The mouse experimental process using a treadmill can be applied in various ways depending on the disease. The aerobic exercise protocol applied in this study was conducted at a speed of 8.0 m/min for 30 min, according to the previously reported findings [73,74].

### 4.5. Ribonucleic Acid (RNA) Isolation, Gene Expression Profiling, and Data Analysis

Total gene expression analysis was performed to determine the effect of radiation when combined with aerobic exercise. Copy DNA libraries were constructed from extracted RNA using the TruSeq Stranded messenger RNA (mRNA) LT sample preparation kit (Illumina, San Diego, CA, USA) according to the manufacturer’s instructions. From the extracted total RNA, only mRNA was used for analysis. The protocol consisted of polyA-selected RNA extraction, RNA fragmentation, random hexamer-primed reverse transcription, and 100 nt paired-end sequencing using an Illumina NovaSeq 6000. Libraries were quantified using quantitative polymerase chain reaction (qPCR), according to the qPCR Quantification Protocol Guide, and qualified using an Agilent Technologies 2100 Bioanalyzer (Aligent Technologies, Santa Clara, CA, USA).

Before analysis, the raw reads were processed from the sequencer to remove low-quality and adapter sequences, and the processed reads were aligned to the *Homo sapiens* genome (Genome Reference Consortium Human Build 38) using Bowtie2 v2.3.4.1. The reference genome sequence and annotation data were downloaded from Ensemble. Transcript assembly was then performed using Feature Counts v1.6.0. Based on the results, transcript and gene expression levels were calculated as read counts or Fragments Per Kilobase of exon per million fragments mapped per sample. Expression profiles were used to perform additional analyses, such as DEG expression analysis. The DEGs that exhibited at least a two-fold change were identified. For DEGs, GO and KEGG were applied with cluster Profiler v3.18.1 in R v4.0.3, which supports the statistical analysis and visualization of functional profiles for genes and gene clusters. Signaling pathway analysis was performed with KEGG to identify the functions of gene sets in specific pathways.

### 4.6. Statistical Analysis

All numerical data were presented as the mean ± standard error of at least three independent experiments. For quantification, data were analyzed using t-tests or analysis of variance. The Prism 5 software (GraphPad Software, San Diego, CA, USA) was used for all statistical analyses. Statistical significance was set at *p* < 0.05. Quantitative immunostaining was performed using tissue slide imaging. Positive cell detection was determined using the image, and analysis was performed by specifying a threshold value (default value: 0.2–0.6) for the designated positive cell detection.

## Figures and Tables

**Figure 1 ijms-25-02757-f001:**
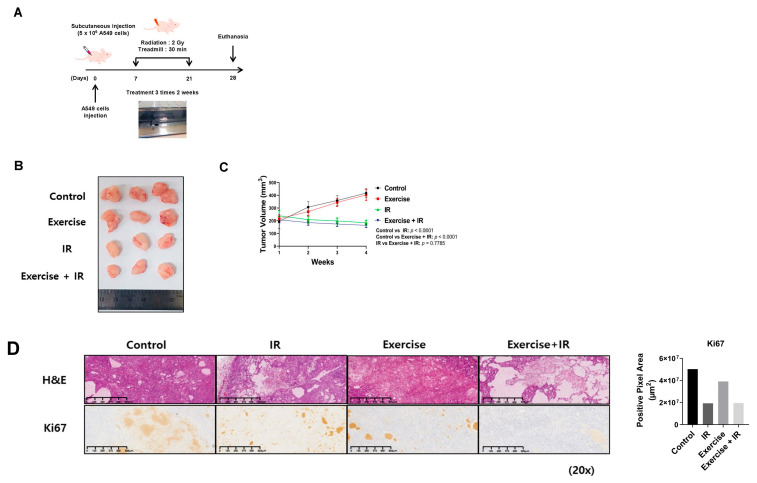
Aerobic exercise in combination with radiation reduced NSCLC growth. (**A**) Experimental procedure in a mouse xenograft model designed to investigate tumor growth in vivo. After subcutaneous injection of A549 cells, BALB/c athymic nude mice (n = 5 per group) were irradiated three times a week for 2 weeks and aerobic exercise was performed for 30 min three times per week; (**B**,**C**) Inhibitory effect on tumor volume in mouse xenograft models subjected to aerobic exercise and radiation; (**D**) H & E staining and Ki-67 immunohistochemical staining of mouse tumor sections showing inhibited cell proliferation upon radiation combined with aerobic exercise.; con: control, Ex: exercise, and IR: irradiation.

**Figure 2 ijms-25-02757-f002:**
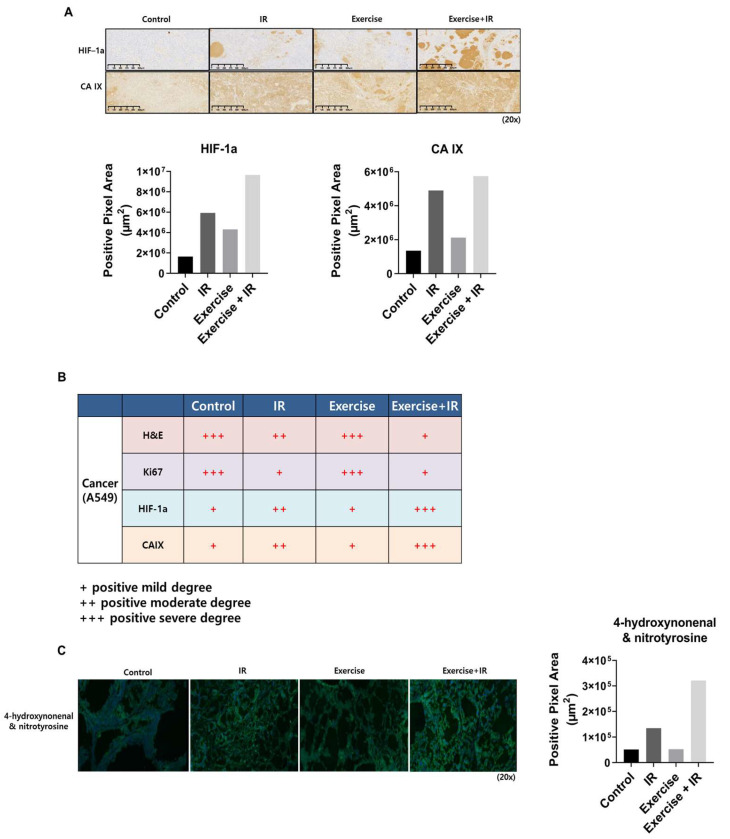
The “oxygen effect” was induced in tumor cells when aerobic exercise was combined with radiation. (**A**,**B**) HIF-α and CA IX immunohistochemical staining of tumor segments from mice treated with aerobic exercise and radiation exhibit increased oxygenation levels in tumor cells; (**C**) increased ROS levels in A549 cells in response to aerobic exercise and radiation confirmed by 4-hydroxynonenal and nitrotyrosine immunohistochemistry.

**Figure 3 ijms-25-02757-f003:**
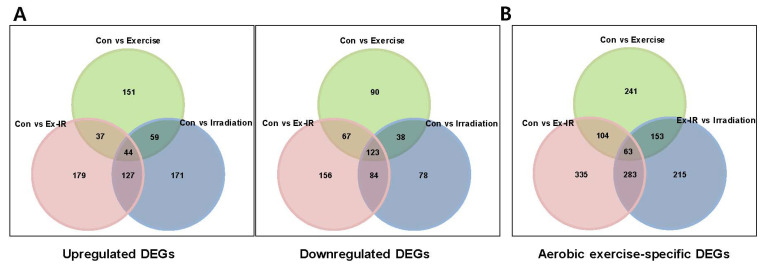
Combining aerobic exercise and radiation induces changes in various genes. (**A**) Through comparative combination analysis, DEGs that were upregulated or downregulated by more than 2-fold compared to the control group were identified. (**B**) Comparisons between the radiation-alone and combination groups: 215 DEGs were specifically altered by aerobic exercise. (**C**) Top 20 major enrichment biological processes in DEGs. The x-axis represents GeneRadio and False Discovery Rate (FDR) values. The y-axis shows various biological processes. (**D**) Chart showing the top 20 major enrichment cellular components of DEGs. The x-axis represents GeneRadio and FDR values. The y-axis shows the various cellular components. (**E**) Chart showing the top 20 Kyoto Encyclopedia of Genes and Genomes (KEGG) signal pathways of DEGs.

**Figure 4 ijms-25-02757-f004:**
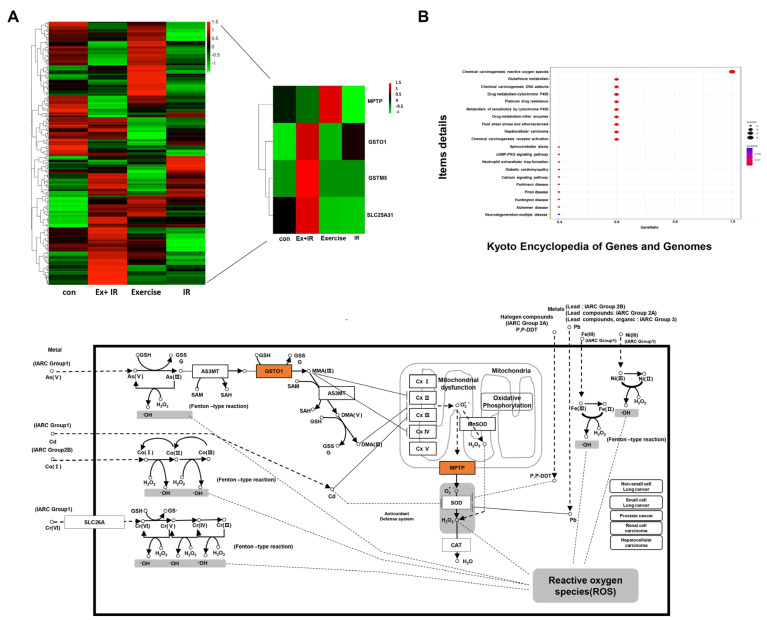
DEGs associated with oxygenation and ROS in the combination group compared to the radiation-alone group. (**A**) Overall heat-map of DEGs associated with oxygenation and ROS, and heat-map of specific genes; (**B**) KEGG analysis focusing on the ROS signaling pathway of DEGs altered in the combination group compared to the radiation-alone group.

**Table 1 ijms-25-02757-t001:** Top 20 DEGs changed when aerobic exercise and radiation therapy were combined.

	Symbol	Fold Change (log2)	Up/DownRegulated
**100422859**	Mir3136	−14.56455009	Downregulated
**100189415**	Trf-Gaa1-3	−13.88255615	Downregulated
**113218489**	miR10398	13.86087595	Upregulated
**406910**	miR125a	−13.83876398	Downregulated
**407011**	miR23b	−13.66512849	Downregulated
**102466194**	miR6773	−13.64055237	Downregulated
**113218510**	miR10523	−13.60208128	Downregulated
**102466741**	miR6813	−13.45770358	Downregulated
**692195**	Snord75	−13.35817707	Downregulated
**26809**	Snord42a	−13.31087593	Downregulated
**442907**	miR339	−13.29544742	Downregulated
**100616294**	miR4740	−13.28779461	Downregulated
**100033413**	Snord116-1	−13.25012801	Downregulated
**100189284**	Trg-Gcc1-5	−13.11534573	Downregulated
**100500904**	miR3942	−13.08187484	Downregulated
**100188995**	Trk-Ctt6-1	−13.07527287	Downregulated
**102465842**	miR7856	−13.04270882	Downregulated
**406900**	miR106b	13.01436327	Upregulated
**102465429**	miR6511b1	12.96253071	Upregulated
**102466743**	miR6822	−12.91934167	Downregulated

## Data Availability

Our whole transcriptome sequencing analysis data are registered in the NCBI database (GEO number: GSE221562).

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
