# Peer review of "Aerobic Exercise Improves Radiation Therapy Efficacy in Non-Small Cell Lung Cancer: Preclinical Study Using a Xenograft Mouse Model"

_ijms, 2024, doi:10.3390/ijms25052757_

Round 1
Reviewer 1 Report
Comments and Suggestions for Authors
WELL DONE,CONGRATS!A VERY NICE APPROACH WITH THE ONLY HANDICAP OF NOT BE EASILY PRACTISED BY LUNG CANCER PATIENTS(YOU WROTE IT TOO).FOR THAT REASON I THINK YOU SHOULD CONSIDER THE OPTION TO BE MORE DETAILED WITH THE ALTERNATIVE METHODS(NANOPARTICLES ETC).FURTHERMORE SEE IF :1) METABOLITES OF AEROBIC CYCLE INTERFEAR IN THE PROCESS,2)FREE RADICALS SHOULD CONTROL THE DURATION OF THE EXERCISE

Author Response
We thank you and the reviewers for your thoughtful suggestions and feedback. The manuscript has benefited from these insightful comments.
The manuscript has been rechecked and the necessary changes have been made in accordance with the reviewers’ suggestions. The responses to all comments have been prepared and attached herewith. All revisions in the manuscript have been indicated with red font for your convenience.

Reviewer 2 Report
Comments and Suggestions for Authors
In this manuscript, Jo and colleagues demonstrated that aerobic exercise amproves radiotherapy efficacy in a xenograft mouse model of non-small cell lung cancer.
The article is well wirtten, it flows straight forward and logically.
Below some my comments.
-en the abstract, it is not clear that a xenograft model has been used
-the treatemnt schedule needs to be clarified. Do the mice first receive radiation therapy and then exercise or vice versa? In the same day?
-line 70-82, that part is out of topic as nano-particles shuttling oxigen are not used in this work
-” In addition, the reduction in tumor volume in the combination group tended to be greater than that in the radiation alone group (Figure 1B, C).” PPease rephrase this sentence as there are no significant differences between the two groups
-” There was a significant difference in H&E staining between the radiation alone group and the combination group”. What do the Authors refer at? Cell composition, tissue archutecture, stroma?
- the Ki67 quantification in figure 1D what does it report? Is it a mean? How many samples? How many image fields?
-same for the plots in Figure 2
- “whole transcriptome analysis was conducted using tissues obtained from the in vivo experiments” Does it mean that the RNA was extracted from tumor tissu? Are these samples includes tumor, stromal and immune cells? Or is it only RNA from tumor cells isolated from tumor tissue?
Author Response

(The authors gave the same response as above.)

Reviewer 3 Report
Comments and Suggestions for Authors
Dear authors,
your article is interesting, but there are some notable concerns that warrant attention and resolution.
1. Figure 1A and C: The authors indicated irradiation when tumor volumes reached 200mm, with mice euthanized after 42 days. However, Figure 1C appears somewhat confusing. It is crucial to clarify whether the figure depicts tumor volumes post-irradiation or at the moment of irradiation. Additionally, the authors measured volumes up to the 4th week, but it remains unclear from which specific starting point this duration is calculated.
2. Furthermore, it raises the question of why the measurements did not extend until the euthanasia of the mice.
3. The correlation of tumor volumes between the IR and IR+exercise groups lacks statistical significance. Consequently, I believe the title fails to accurately reflect the content of this article.
4. Were similar experiments conducted using a second cell cultures?
5. I recommend enhancing the quality of the analysis of immunohistochemistry (IHC) images.
6. Concerning HIF-1a, is the observed increase in HIF-1a levels a result of factor stabilization or gene overexpression? It is advisable to conduct quantitative PCR (qPCR) experiments for a more comprehensive understanding. Additionally, previous studies have associated elevated HIF-1a levels with resistance to radiation. How do the authors reconcile these contradictory findings in their explanations?
Kind regards
Round 2
Reviewer 2 Report
Comments and Suggestions for Authors
The manuscript by Jo and colleagues is improved after revision.
The Authors replied point-by-point to the reviewers’comments. However, there is still a concern:
-Ki67 quantification: the authors must clarify what the plots represent. The addition included in the revised manucript does not fullfil this comment
Author Response
Response to Reviewer 2 Comments 1. Summary Thank you for taking the time to review our manuscript. Please find our detailed responses below and the corresponding revisions/corrections in red font or indicated with "track changes" in the re-submitted files. 2. Point-by-point response to Comments and Suggestions for Authors Comments 1: -Ki67 quantification: the authors must clarify what the plots represent. The addition included in the revised manucript does not fullfil this comment. Response 1: Thank you for highlighting this. As per your suggestion, the meaning of the results of H&E and Ki-67 staining has now been specified in the revised manuscript. (page 3, line106-109)

Reviewer 3 Report
Comments and Suggestions for Authors
Dear authors,
Thank you for your response to my question.
In my opinion the lack of statistical significance and the absence of similar findings in at least one more cell line constiute a significant limitation for this study.
Kind regards,
Round 3
Reviewer 3 Report
Comments and Suggestions for Authors
Dear Authors,
Thank you for your prompt response. I appreciate the adjustments made to the paper format; however, I still have lingering concerns regarding the substance of the research.
While the changes addressed some aspects, such as formatting, they failed to address critical inquiries I raised regarding the methodology and scope of the study. For instance, the rationale behind the use of a single cell line and the omission of PCR experiments remains unclear. Additionally, the absence of tumor measurements until euthanasia presents a significant gap in data collection.
Considering these issues, I acknowledge the potential significance of your findings but must emphasize that they appear preliminary. To validate and strengthen your observations, I recommend conducting supplementary experiments.
I look forward to your further insights on these matters.
Round 4
Reviewer 3 Report
Comments and Suggestions for Authors
Dear authors,
thank your response and your improvements for this paper.
Kind regards